# Comprehensive Investigations on Fluid Flow and Cavitation Characteristics in Rotating Disk System

**Junyu Sun** [1,*] , **Liyu Chen** [2] **, Hua Huang** [1] **, Bing Zhang** [1] **and Pengfei Qian** [1,3]

1 School of Mechanical Engineering, Jiangsu University, Zhenjiang 212013, China
2 School of Mathematics and Computational Science, Tangshan Normal University, Tangshan 063000, China
3 The State Key Laboratory of Fluid Power and Mechatronic Systems, Zhejiang University, Hangzhou 310027, China
* Correspondence: sunjunyuit@outlook.com

**Featured Application: The findings can be used to prevent or predict cavitation in hydraulic machinery, such as hydro-viscous clutches, wet clutches, disk turbines, and disk motor cooling systems, which can improve lubrication performance and strengthen dynamic pressure bearing capacity between rotating disk systems.**

**Abstract:** The present work numerically investigates the flow behavior of Newtonian fluid between rotating disk systems. Rotating disk flow is an embranchment of fundamental flow research, which forms the theoretical basis for the flow media in many disk-type hydraulic machinery, and has been widely applied in energy, aerospace, vehicles, medical, and chemical areas. The Reynolds equation model of the rotating disk flow is established based on a series of assumptions, and considers the characteristics of hexahedral surface textures on the friction disk, and the control equation is dimensionless. The velocity, pressure, and pressure coefficient in oil film are solved by finite difference algorithm, and the shear cavitation incipient position is predicted. Graphical visualization and numerical data about cavitation characteristics are also presented. Furthermore, the cavitation process is researched in CFX and the experiment, eventually verifying the correctness of the numerical analysis. The pressure perturbation caused by texture boundaries and the low pressure at the vortex center induce the initial generation of cavitation, and cavitation effect can obviously enhance lubricating performance. The cavitation region extends from micro-dimples towards non-textured region when texture rate increases. This study can guide design of surface structures such as mechanical seals, wet clutches, and disk turbines.

**Keywords:** hydraulic machinery; cavitation prediction; hydrodynamic lubrication; surface texture

## 1. Introduction

In wet clutch, hydro-viscous clutch, and other transmission components and mechanical seals, the complex force interaction between oil and disk structures directly affects the transmission efficiency of the vehicle transmission system and the reliability of the seal. Firstly, Karman [1] studied the flow on rotating disk in 1921, and discussed the rotating flow of a stationary fluid on the surface of an infinitely large rotating disk, i.e., the "free disk" problem, which is actually a common fluid flow problem in rotating equipment such as compressors and centrifugal pumps. Based on the assumption of axisymmetry and similar transformations, he first transformed the control equations for constant incompressible axisymmetric flow in cylindrical coordinates into a set of nonlinear differential equations and found that the angular velocity of the fluid decreases gradually from the rotational velocity of the disk to zero as the axial distance from the disk increases. The fluid boundary layer attached to the disk surface was called the "Von Karman boundary layer" or "Ekman layer". Cochran [2] used numerical integration to solve the system of equations proposed

by Karman to obtain the axial and radial velocity distributions of the fluid in the "free disk" problem. Cochran [2] used numerical integration to solve the system of equations proposed by Karman to obtain the axial and radial velocity distributions of the fluid in the "free disk" problem. Based on Karman's analysis, Bodewadt [3] studied the flow formed by rotating an infinitely large stationary disk with constant angular velocity. In this case, the angular velocity of fluid gradually decreases from the constant angular velocity to zero as it approaches the disk surface, and the boundary layer adhering to the disk surface is called the "Bodewadt boundary layer". With the development of fluid drive and control technology, people have a new understanding of disk flow [4–6]. Klingl [7] analyzed whether the rotating disk boundary layer can still guarantee absolute stability when disk system spacing is sufficiently small, and the results showed convective instability when radial flow is also present. Vijay [8] studied the flow behavior of magnetic fluids in rotating interfaces and found that the velocity near the passive sheet decreased when the viscosity increased, and fluid temperature as well as the concentration and Reynolds number were inversely proportional. Lin [9] et al. used four dimensionless cases to analyze the strongly rotating flow and the stability in the turbine disk cavity and found that the parameters such as flow Reynolds number, rotational Mach number, and rotational Reynolds number affect the stability of the disk cavity system greatly.

Surface micro-texture technology refers to the introduction of structures, such as micro-dimples, micro-grooves, micro-convex bodies, etc., to improve the frictional lubrication properties of the sliding mechanical interface [10–14]. Hamilton [15] et al. were the pioneers to analyze the effect of micromorphology between parallel plates on the dynamic pressure bearing capacity through the dimensionless and similarity principle of equation, and the effect of factors such as depth, arrangement distribution and relative area of the micromorphology. Siripuram and Stephens [16] researched the effect of different texture dimples on hydrodynamic lubrication and friction performance and found that it affected the frictional coefficient lightly, and triangular texture can significantly increase load carrying capacity reducing leakage of oil. Yu [17,18] et al. analyzed the effect of texture shape and texture orientation on hydrodynamic pressure by a super-relaxation iterative algorithm and used average dimensionless pressure as an index to assess bearing capacity, and found that elliptical dimples perpendicular to the sliding direction produced the best lubrication performance. Liang [19] et al. explored the relationship between texture position, texture depth, and oil-film-bearing capacity and found that the bearing capacity could be improved in both converging and diverging regions, and shallow texture in converging regions could better improve lubrication performance, while effect in diverging region was not significant. Han [20] et al. modeled the spherical crown texture and simplified it dimensionless, simulated the 3D model in Fluent, and found that the increase of Reynolds number and texture depth enhances the fluid-bearing capacity. Etsion [21] nondimensionalized the Reynolds equation and established the hydrodynamic-pressure lubrication equation for a non-contact mechanical seal with hemispherical texture and found that as dimple diameter decreases, the oil-bearing capacity and leakage decreases, the torque increases, and the optimal dimple size is related to the viscosity of the oil. Tala-Ighil [22] et al. used the finite difference method to simulate the effect of columnar texture on dynamic pressure radial bearings, based on analysis of different texture arrangements, and found that the bearing surface texture rate of 100% was not effective in increasing the dynamic bearing capacity, and the best effect was achieved when the attitude angle of the texture was 49°, and the thickness of the oil film was able to increase by 1.8%. Arif [23] et al. investigated the effect of texture rate on sliding bearings and found the stability of bearing operation decreases when the texture rate increases to a certain value; the oil temperature increases by 10% after textured compared to the untextured bearing.

Cavitation is specific and occurs in a low pressure region inside liquid, especially at the interface between liquid and solid. Conventionally, cavitation is a phase change from liquid [24–27]. As shown in Figure 1, the morphological characteristics of the friction pairs severely affect the intensity of shear cavitation [28]. Ma [29] pointed out that when

the aircraft is flying in rain or clouds, the edges of the turbofan blades in turbine are affected by cavitation causing surface material loss, which increases the surface roughness of turbofan blades and reduces the efficiency of the aircraft flight. Jahangir [30] studied the corrosion effect of fluid cavitation on material surfaces by controlled experiments. A pressure difference was applied in flow field and significant vortices in the flow field with a blunt body was caught by high-speed camera. Through continuous experiments for up to 30 h, it was found that extensive erosion occurred in the flow field with a blunt body. Nagalingam [31] added abrasives into fluid to finish the internal surfaces of mechanical parts. It was found that tiny abrasive particles increased the cavitation erosion effect and increased the cleaning degree by 90%. Nguyen [32] et al. studied the underwater vehicle cavitation and analyzed the special flow due to cavitation bubbles being created, dislodged, and collapsed, which provides guidance for the design of underwater vehicles to prevent cavitation. Kadivar [33] studied the effect of microstructured surfaces on cavitation to reduce the hazard level of cavitation of ship propellers and found that the momentum of microjet flow produced by cavitation bubbles ruptured on microstructured surfaces and smooth surfaces was different, and microstructured surfaces can effectively reduce the annular ring after bubbles rupture, while surfaces machined with microporosity can reduce the cavitation area. Nowakowska [34] analyzed the preventive effect of various coatings on cavitation and found that cavitation always starts at pores or small cracks on the surface of the object, leading to brittle damage of the material as a whole and eventually to large cavitation dimples.

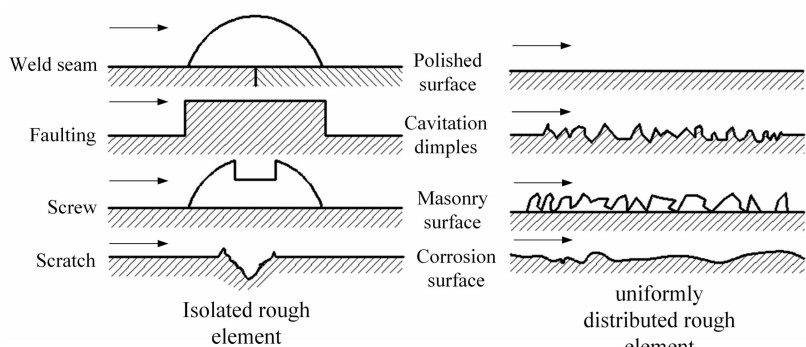

**Figure 1.** Irregular surface of flow field boundary (The arrow points in the direction of fluid flow).

Based on the summary of the literature, research on hydromechanical cavitation has mainly focused on the groove structure, while less research has been carried out on the cavitation of tiny surface structures, especially on the mechanism of cavitation formation by micro-textures. In our research, Reynolds equation model of rotating disk flow is developed, and the control equation is dimensionless processed to predict cavitation incipient position on rotating textured disk, and the predicted results are shown using graphs and data. The accuracy of the numerical analysis is further verified by CFX simulation. This paper also proposes specific methods to reduce cavitation effect, which can provide guidance for the design of contact surface structures such as wet clutches, liquid viscous clutches, dynamic bearings, and mechanical seals with practical engineering applications.

## 2. Numerical Analysis

### 2.1. Cavitation Inception Mechanism

#### 2.1.1. Pressure Coefficient

Microscopically, cavitation inception is the process by which molecules with higher kinetic energy in liquid overcome the gravitational force of liquid surface and escape from the liquid. The key impetus for cavitation is the formation of the cavitation nucleus, and

only low pressure of a sufficient duration will cause cavitation. Thomas described the phenomenon by a dimensionless quantity $\sigma$, the cavitation number [35]:

$$\sigma = \frac{p_\infty - p_v}{\frac{1}{2}\rho u^2_\infty} \tag{1}$$

where $p_\infty$ is the static pressure of the fluid in undisturbed reference section, in Pa; $u_\infty$ is the fluid velocity undisturbed reference section, in m/s; $\rho$ is the fluid density, in kg/m$^3$. $p_v$ is the fluid saturated vapor pressure, in Pa. The cavitation number can quantitatively describe the cavitation phenomenon, and from Equation (1) that the cavitation number is mainly determined by the pressure and fluid velocity and is related to the material parameters.

### 2.1.2. Critical Pressure of Cavitation Inception

The cavitation number $\sigma$ can describe the cavitation phenomenon quantitatively. Set $\sigma_{crit}$ as the critical cavitation number when cavitation has just arisen or just disappeared, no cavitation when $\sigma > \sigma_{crit}$; when $\sigma = \sigma_{crit}$ cavitation has just occurred or just disappeared; When $\sigma \ll \sigma_{crit}$ cavitation is fully developed, even super cavitation may appear. In Equation (1) $p_\infty$ and $u_\infty$ are the pressure and velocity in the far-field undisturbed reference section, and the pressure coefficient $C_p(x)$ is obtained by introducing the position-of-concern pressure $p_x$ and the position-of-concern velocity $u_x$ [36].

$$C_p(x) = \frac{p_x - p_\infty}{\frac{1}{2}\rho u^2_x} \tag{2}$$

where $p_x$ is the position-of-concern pressure, Pa; $u_x$ is the position-of-concern velocity, m/s; Let $p_x - p_\infty = \Delta p$, i.e., $\Delta p$ is pressure difference between the position-of-concern and reference position, Pa.

Comparing the expressions of cavitation number $\sigma$ and pressure coefficient $C_p(x)$, we can see that $\sigma = C_p(x)$, when the pressure at position $x$ is the smallest, i.e., $p_x = p_{min}$, cavitation will occur first at position $x$.

### 2.1.3. Effect of Surface Structure on Cavitation

The pressure change is an external factor of cavitation occurrence, and the local pressure perturbation caused by the rough surface, which generates the low-pressure area, provides the external conditions for cavitation. We introduced various types of rough surfaces in the above. Assuming that there is an unevenness at $F$ position of the smooth surface, which causes a perturbation of the local pressure $p_1$ as $p_{ml}$:

$$\sigma = \frac{p_\infty - p_v}{\frac{1}{2}\rho u^2_\infty} = \frac{p_\infty - p_1}{\frac{1}{2}\rho u^2_\infty} + \frac{p_{ml} - p_v}{\frac{1}{2}\rho u^2_\infty} + \frac{p_1 - p_{ml}}{\frac{1}{2}\rho u^2}\left(\frac{u}{u_\infty}\right)^2 \tag{3}$$

where $u$ is the local velocity at location $F$. The condition for cavitation at an uneven local location is $p_{ml} = p_v$.

Isolated rough elements only disturb fluid pressure in their vicinity and have negligible effect on the overall pressure distribution in the flow field, while uniformly distributed rough elements increase the intensity of fluid turbulence. Arndt [23] deduced from turbulent flow theory that:

$$\sigma_{11} = K\frac{\tau_0}{\frac{1}{2}\mu u^2_\infty} \tag{4}$$

where $\tau_0$ is shear stress, Pa; the value of the coefficient $K$ is determined by the test, according to experience $K = 16$ is the most suitable value.

### 2.1.4. Methods for Predicting Cavitation

Early studies on cavitation focused on experimental observations and theoretical studies. Theoretical research methods include bubble dynamics forecasting methods,

fuzzy forecasting methods, and pressure field forecasting methods [37]. The pressure field forecasting method is based on pressure changes in fluid. According to the previous analysis, cavitation occurs first at the lowest pressure position in the flow field. Therefore, the flow field pressure coefficient $C_p(x)$ is calculated using experimental measurements or hydrodynamic methods and the minimum pressure coefficient $C_{pmin}$ is found as follows:

$$C_p(x) = \frac{p_x - p_\infty}{\frac{1}{2}\rho u_\infty^2} \leq \frac{p_v - p_\infty}{\frac{1}{2}\rho u_\infty^2} = C_{P_{\min}} \tag{5}$$

The relationship between the cavitation number and the minimum pressure coefficient is: $\sigma_i = -C_{pmin}$. Additionally, this method is relatively small in calculation, so the pressure field forecasting method is used to predict the location of cavitation inception.

*2.2. Reynolds Equation and Its Dimensionlessification*

2.2.1. Reynolds Equation

The existing literature contains a large amount of cavitation inception data with water as the medium, our research is to analyze the shear cavitation phenomenon of the flow field between a pair of rotating friction disks, and the simplified model is shown in Figure 2.

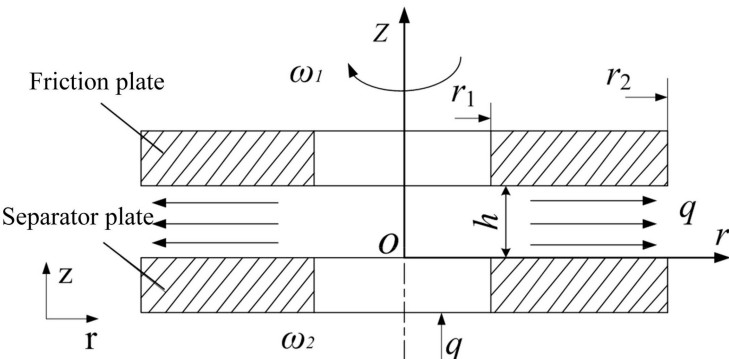

**Figure 2.** Simplified model for friction pair ($\omega_1$ is rotation speed of friction plate; $\omega_2$ is rotation speed of separator plate; $r_1$ is inner radius; $r_2$ is outer radius; $h$ is oil film thickness; $q$ and nearby arrows are directions of oil flow).

Because the working medium is chosen to be a viscous hydraulic fluid, the calculation is simplified assuming that the fluid is a Newtonian fluid and the viscosity is constant. The following assumptions are made:

(1) No relative slippage on the contact surface of fluid oil and friction disk;
(2) The disks are always parallel to each other without bias grinding, the oil film thickness is always greater than 0, and there is no convex peak contact;
(3) Flow field flow is laminar flow;
(4) The effect of volume force and inertia terms are neglected and only the effect of centrifugal terms is considered;
(5) The axial movement of oil is neglected.

Based on the above assumptions the Reynolds equation for the flow of oil between frictional pair can be obtained [38,39]:

$$\frac{\partial}{\partial r}\left(rh^3\frac{\partial p}{\partial r}\right) + \frac{1}{r}\frac{\partial}{\partial \theta}\left(h^3\frac{\partial p}{\partial \theta}\right) = 6\mu\omega r\frac{\partial(h)}{\partial \theta} + \frac{3}{10}\rho_L\omega^2\frac{\partial(r^2h^3)}{\partial r} \tag{6}$$

where $p$ is pressure, Pa; $\mu$ is oil dynamic viscosity, Pa·s; $h$ is oil film thickness, m; $\omega$ is angular velocity, rad/s. To make Equation (6) dimensionless, introduce dimensionless parameters as described in Equation (7):

$$P = \frac{p}{p_0}, \ \Phi = \frac{\theta}{2\pi}, \ H = \frac{h}{h_0}, \ R = \frac{r}{r_2}, \ K_1 = \frac{3\mu\omega r_2^2}{\pi p_0 h_0^2}, \ K_2 = \frac{\pi\rho_L\omega h_0^2}{10\mu} \tag{7}$$

The Reynolds equation was made dimensionless by combining Equation (6), as:

$$\frac{\partial}{\partial R}\left(RH^3\frac{\partial P}{\partial R}\right) + \frac{1}{4\pi^2 R}\frac{\partial}{\partial \Phi}\left(H^3\frac{\partial P}{\partial \Phi}\right) = K_1\left[R\frac{\partial(H)}{\partial \Phi} + K_2\frac{\partial\left(R^2 H^3\right)}{\partial R}\right] \tag{8}$$

Given the periodicity in oil film, the sector-shaped oil film with angle $\alpha$ is selected for the solution, and the dimensionless boundary condition of Equation (8) is:

$$\begin{cases} P|_{R=r_1/r_2} = 1, \ P|_{R=1} = 0 \\ P|_{\Phi=0} = P|_{\Phi=\alpha/2\pi}, \ \frac{dP}{d\Phi}\Big|_{\Phi=0} = \frac{dP}{d\Phi}|_{\Phi=\alpha/2\pi} \end{cases} \tag{9}$$

The friction disk rotates fast, and the velocity in the flow field can be viewed as the synthesis of radial and circumferential velocities, and the dimensionless velocity components are as follows [40]:

$$\begin{cases} u^* = u/\frac{p_0 h_0^2}{8\mu r_0} = -H^2\left(\frac{\partial P}{\partial R} - K_1 K_2 R\right) \\ v^* = v/\frac{p_0 h_0^2}{16\pi\mu r_0} = -\frac{H^2}{R}\frac{\partial P}{\partial \Phi} + 4\pi^2 K_1 R \end{cases} \tag{10}$$

### 2.2.2. Texture Distribution and Oil Film Thickness

Figure 3 has shown the schematic diagram of homogeneous hexahedral texture, $\alpha$ is center angle of oil film, the distribution of textures has $N$ columns along the radial direction, each column is $M$, the texture number is $N \times M$, the central angle of texture is $\alpha_1$, the texture depth is $h_t$.

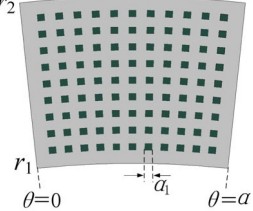

**Figure 3.** Uniform distribution of hexahedron textures ($r_1$ is inner radius; $r_2$ is outer radius; $\alpha_1$ is angle of a texture structure; $\theta$ is center angle; The dark area is textured area; The light-colored area is the non-textured area).

Then, the right boundary $\Phi_r$ and left boundary $\Phi_l$ of the texture structure can be expressed, respectively, as:

$$\begin{cases} \Phi_r = \frac{\alpha}{2\pi(N+1)} - \frac{\alpha_1}{4\pi} \\ \Phi_l = \Phi_r + \frac{\alpha_1}{2\pi} \end{cases} \tag{11}$$

Based on texture distribution mentioned above, if we set the radial length of the textures as $L$, and the distance between adjacent textures in the radial direction as $\Delta m$, then $\Delta m = (r_2 - r_1)/r_2/(M + 1)$. If we set the circumferential angle between every two adjacent textures as $\Delta\alpha$, then, $\Delta\alpha = (\alpha/2\pi)/(N + 1)$. The radial lower boundary of the texture is a circular arc $C_{R1}$ with radius $R_1 = r_1/r_2 + n\Delta m + (n - 1)L/r_2$, and the upper boundary is a circular arc $C_{R2}$ with radius $R_2 = r_1/r_2 + n\Delta m + nL/r_2$, where $n$ denotes the $n^{th}$ texture along the inner radius toward outer radius.

After defining the circumferential left and right boundaries $\Phi_l$ and $\Phi_r$ and the radial upper and lower boundaries $C_{R1}$ and $C_{R2}$ of the texture dimple, the oil film can be divided into two parts: the texture area and the non-texture area. When $R_1 \leq R \leq R_2$, and $\Phi_r + k\Delta\alpha \leq \Phi \leq \Phi_l + k\Delta\alpha$, the thickness of oil film is $h = h_t + h_0$. When $R \leq R_1$ or $R_2 \leq R$ and $k\Delta\alpha \leq \Phi \leq \Phi_{gb} + k\Delta\alpha$ or $\Phi_{gt} + k\Delta\alpha \leq \Phi \leq (k+1)\Delta\alpha$, the thickness is $h_0$, $n = 1, 2, 3 \ldots M$, $K = 1, 2, 3 \ldots N - 1$.

When $h_0 = 0.1$ mm, $\alpha = 12°$, $h_t = 0.05$ mm, the oil film thickness is shown in Figure 4.

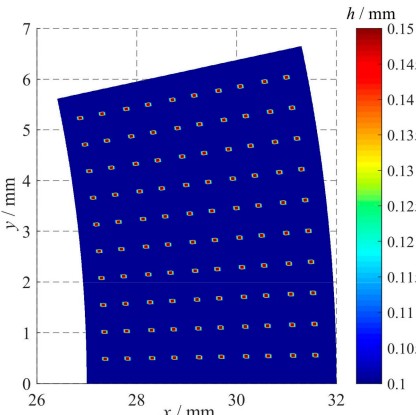

**Figure 4.** Film thickness under uniformly distributed hexahedron texture.

### 2.3. Reynolds Equation Dimensionlessification

As shown in Figure 5, the basic idea of the finite difference algorithm is to replace the continuous fixed solution region with a grid composed of a finite number of discrete points, which are used as nodes of the grid. For the oil film discussed above, the *r* direction is divided into m cells and the $\theta$ direction is uniformly divided into *n* cells, then the radial and circumferential steps of the oil film are $\Delta r = 0.005/m$ and $\Delta\theta = 12/n$, respectively.

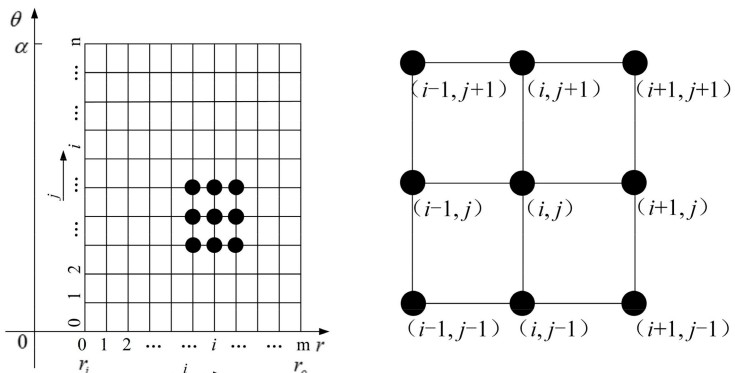

**Figure 5.** Solution domain discretization of finite difference algorithm (The black circles are adjacent discrete points.).

Differentiate the dimensionless Reynolds equation in the discrete solution domain.

$$H^3 \frac{\partial P}{\partial R} + R\frac{\partial}{\partial R}\left(H^3 \frac{\partial P}{\partial R}\right) + \frac{1}{4\pi^2 R}\frac{\partial}{\partial \Phi}\left(H^3 \frac{\partial P}{\partial \Phi}\right)$$
$$= K_1\left[R\frac{\partial(H)}{\partial \Phi} + K_2\left(2RH^3 + R^2\frac{\partial(H^3)}{\partial R}\right)\right] \tag{12}$$

where the first-order partial derivatives and second-order partial derivatives are denoted as:

$$\begin{cases} \frac{\partial P}{\partial R} = \frac{P_{(i+1,j)} - P_{(i-1,j)}}{2\Delta R} \\ \frac{\partial P}{\partial \Phi} = \frac{P_{(i,j+1)} - P_{(i,j-1)}}{2\Delta \Phi} \end{cases} \tag{13}$$

$$\begin{cases} \frac{\partial}{\partial R}\left(H^3 \frac{\partial P}{\partial R}\right) = \frac{H^3_{(i+1/2,j)}P_{(i+1,j)}+H^3_{(i-1/2,j)}P_{(i-1,j)}-\left(H^3_{(i+1/2,j)}+H^3_{(i-1/2,j)}\right)P_{(i,j)}}{(\Delta R)^2} \\ \frac{d}{d\Phi}\left(H^3 \frac{\partial P}{\partial \Phi}\right) = \frac{H^3_{(i,j+1/2)}P_{(i,j+1)}+H^3_{(i,j-1/2)}P_{(i,j-1)}-\left(H^3_{(i,j+1/2)}+H^3_{(i,j-1/2)}\right)P_{(i,j)}}{(\Delta\Phi)^2} \end{cases} \tag{14}$$

Substituting Equations (13) and (14) into Equation (12) can obtain the dimensionless Reynolds equation in the differential form:

$$A_{(i,j)}P_{(i+1,j)} + B_{(i,j)}P_{(i-1,j)} + C_{(i,j)}P_{(i,j+1)} + D_{(i,j)}P_{(i,j-1)} - E_{(i,j)}P_{(i,j)} = F_{(i,j)} \tag{15}$$

where $A_{(i,j)} \sim F_{(i,j)}$ are the discrete point coefficients:

$$A_{(i,j)} = \frac{H^3_{(i,j)}}{2\Delta R} + \frac{R_{(i,j)}H^3_{(i+1/2,j)}}{(\Delta R)^2} \tag{16}$$

$$B_{(i,j)} = -\frac{H^3_{(i,j)}}{2\Delta R} + \frac{R_{(i,j)}H^3_{(i-1/2,j)}}{(\Delta R)^2} \tag{17}$$

$$C_{(i,j)} = \frac{1}{4\pi^2 R_{(i,j)}}\frac{H^3_{(i,j+1/2)}}{(\Delta\Phi)^2} \tag{18}$$

$$D_{(i,j)} = \frac{1}{4\pi^2 R_{(i,j)}}\frac{H^3_{(i,j-1/2)}}{(\Delta\Phi)^2} \tag{19}$$

$$E_{(i,j)} = \frac{R_{(i,j)}\left(H^3_{(i+1/2,j)}+H^3_{(i-1/2,j)}\right)}{(\Delta R)^2} + \frac{H^3_{(i,j+1/2)}+H^3_{(i,j-1/2)}}{4\pi^2 R_{(i,j)}(\Delta\Phi)^2} \tag{20}$$

$$F_{(i,j)} = K_1\left[R_{(i,j)}\frac{H_{(i,j+1)}-H_{(i,j+1)}}{2\Delta\Phi} + K_2\left[2R_{(i,j)}H^3_{(i,j)} + R^2_{(i,j)}\frac{H^3_{(i+1,j)}-H^3_{(i-1,j)}}{2\Delta R}\right]\right] \tag{21}$$

The pressure approximation solution $P_{(i,j)}$ of the Reynolds equation can be expressed as:

$$P^k_{(i,j)} = w\frac{A_{(i,j)}P_{(i+1,j)} + B_{(i,j)}P_{(i-1,j)} + C_{(i,j)}P_{(i,j+1)} + D_{(i,j)}P_{(i,j-1)} - F_{(i,j)}}{E_{(i,j)}} + (1-w)P^{k-1}_{(i,j)} \tag{22}$$

The relevant parameters are solved to calculate the velocity and pressure in the oil film, and the position prediction of cavitation incipient and the corresponding calculation flowchart is shown in Figure 6.

### 2.4. Results of Numerical Analysis

The parameters of the oil film are set in Table 1 when solving oil film using the finite difference algorithm.

**Table 1.** Parameters of flow field.

| Parameters | Values | Parameters | Values |
|---|---|---|---|
| Inner radius $r_1$ (mm) | 27 | Oil film thickness $h_0$ (mm) | 0.1 |
| Outer radius $r_2$ (mm) | 32 | Texture depth $h_t$ (mm) | 0.05 |
| Number of textures $N_1$ | $N \times M$ | Inlet pressure $p_0$ (MPa) | 0.08 |
| Radial number $N$ | 10 | Rotational speed $\omega$ (rpm) | 6000 |
| Circumferential number $M$ | 10 | Oil viscosity at 30 °C $\mu$ (Pa·s) | 0.05 |
| Oil density $\rho$ (kg/m$^3$) | 865 | Oil saturated vapor pressure $p_v$ (Pa) | 1000 |

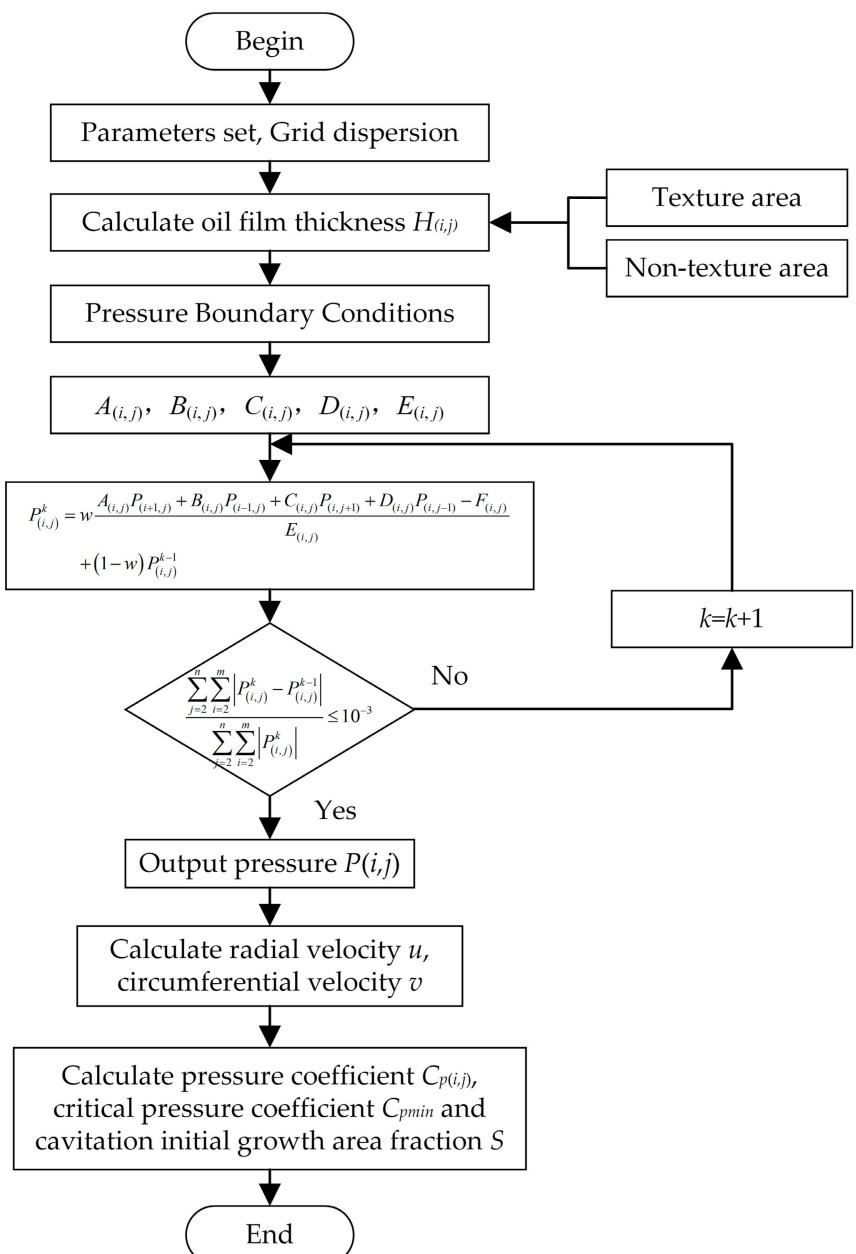

**Figure 6.** Flowchart of finite difference algorithm for Reynolds equation.

### 2.4.1. Effect of Cavitation on Oil Film Velocity

To study the flow field characteristics with texture, non-texture oil film was selected as the control group, with the same condition parameters. Firstly, the solution is performed for the control group. Figure 7 shows the velocity distribution of the non-texture oil film. The radial velocity decreases along the radius direction, when $r = r_1$, $u = 0.8489$; the angular velocity increases along the radius direction, $v = 12.71$ m/s for $r = r_1$ and $v = 15.13$ m/s for $r = r_2$.

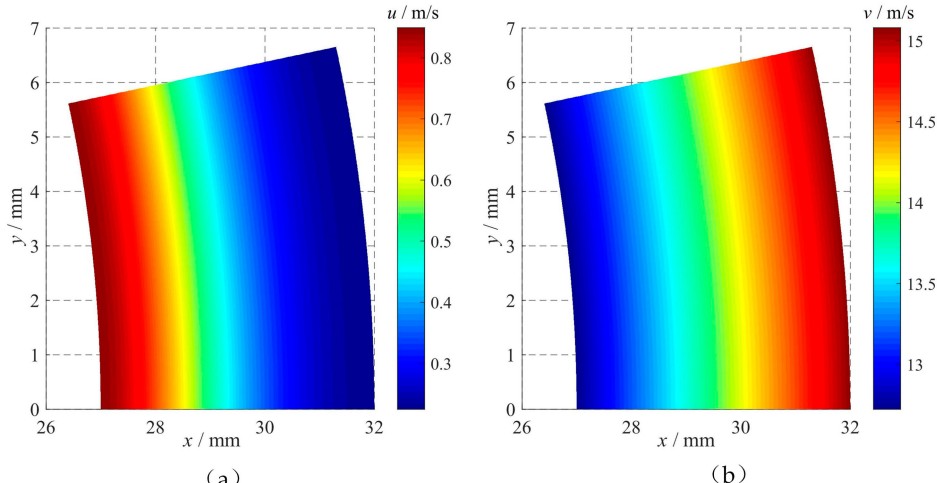

**Figure 7.** Radial velocity and angular velocity distribution of oil film without textures ((**a**) is radial velocity; (**b**) is angular velocity).

Figure 8 shows the velocity distribution of uniform hexahedral textured oil film. Compared with the oil film without texture in Figure 7, the velocity distribution is no longer uniform, the velocity distribution in the non-texture region is basically the same as the velocity in the oil film without textures, and the flow fluctuates more when the oil enters the texture dimples. The minimum radial velocity is −1.89 m/s and the maximum value is 2.71 m/s. The reason is that the oil suddenly increases at the texture entrance and the oil film thickness suddenly decreases at the exit, and the change in film thickness causes the velocity to change drastically. Circumferentially, when the oil enters the texture dimples, the speed decreases rapidly, and the minimum value is 6.1 m/s. It may be that the oil enters into micro-dimples from radial lower boundary, and the vortex causes negative speed and then flows rapidly from the radial upper boundary.

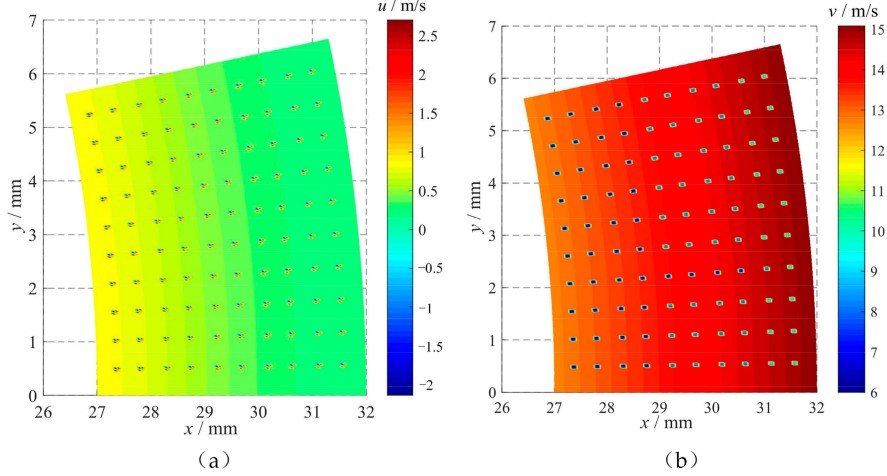

**Figure 8.** Velocity field of oil film with uniformly distributed hexahedron textures ((**a**) is radial velocity; (**b**) is angular velocity).

### 2.4.2. Effect of Cavitation on Oil Film Pressure

The pressure distribution of the oil film without texture and oil film with homogeneous hexahedral textures are compared in Figure 9. In the oil film without texture, pressure is uniform distributed, with pressure values decreasing along the radius direction. The pressure distribution of the oil film with texture is generally consistent with that of the non-textured oil film. The overall effect is not significant, but sudden pressure changes occur in the texture dimples and the local area around them.

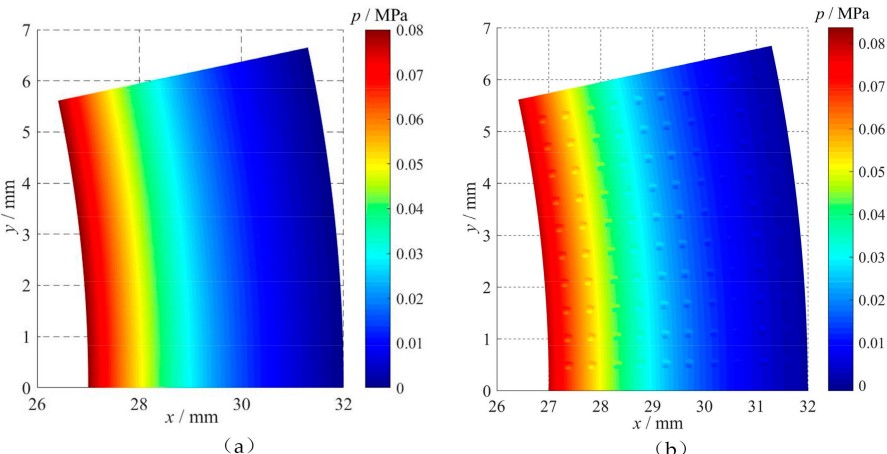

**Figure 9.** Pressure distribution of nontextured and textured oil film ((**a**) is oil film with textures; (**b**) is oil film without textures).

In Figure 10, when the oil is flowing, local high pressure induced by the left boundary of texture extends to the left, and when it passes through the right boundary, a local low pressure appears, and a negative pressure appears in the center of the textures at the outer radius.

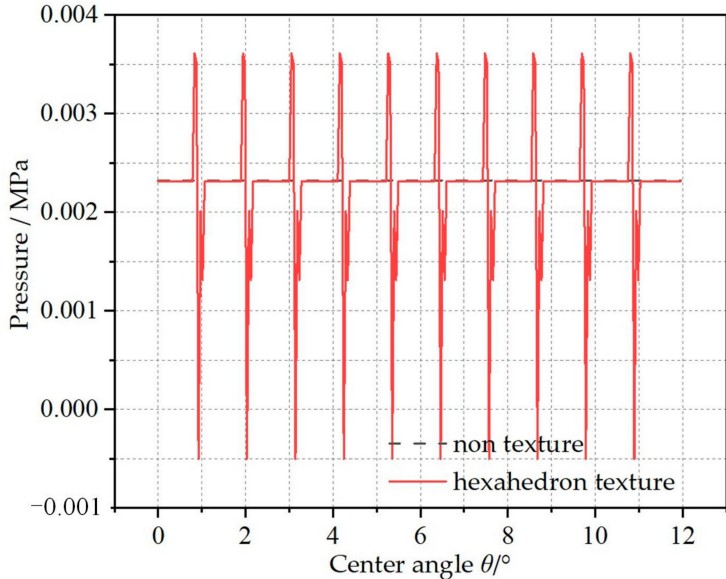

**Figure 10.** Effect of texture on pressure at outer radius.

### 2.4.3. Effect of Cavitation on Pressure Coefficient

According to research above, the texture dimples can induce disturbance to velocity and pressure distribution in the flow field near dimples. The oil velocity without texture disturbance is chosen as the reference velocity u∞ for critical pressure coefficient $C_{pmin}$, the oil velocity with texture disturbance is chosen as the local velocity $u_x$ for the pressure coefficient $C_p(x)$, and the pressure value with texture disturbance is chosen as the local pressure $p_x$ for the pressure coefficient.

Figure 11 shows the distribution of pressure coefficient of the oil film without texture and with homogeneous hexahedral texture. Without textures, the pressure coefficient decreases along the radial direction, the value is greater than the critical pressure coefficient, and the cavitation inception area fraction $S$ is 0, i.e., no cavitation is generated. With textures, the pressure coefficient distribution is generally consistent with that of the oil film

without textures, but in the texture region at the outer radius, the value is lower than $C_{pmin}$, and $S$ is 0.0049.

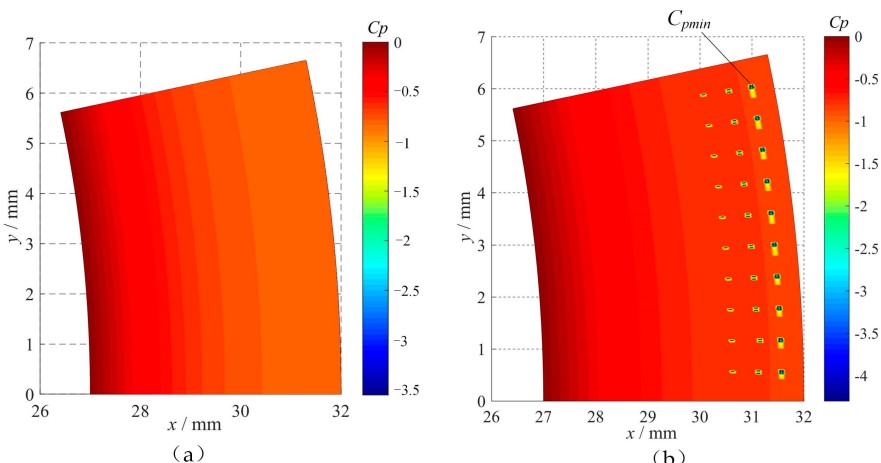

**Figure 11.** Pressure coefficient of nontextured oil film and textured oil film ((**a**) is oil film with textures; (**b**) is oil film without textures).

Isolated rough elements and homogeneous rough elements have distinct separate turbulence on the flow field, and it needs to be confirmed whether texture has the same impact on the flow field pressure at low and high densities. Take $N = 10$ and $M = 20$ to encrypt the texture distribution. When the texture rate is increased, the pressure coefficient at the outer diameter decreases significantly and is lower than the critical pressure coefficient $C_{pmin}$ (Figure 12), the cavitation inception area $S$ increases to 0.102, and it is discovered that the cavitation area increases rapidly when the texture density is increased.

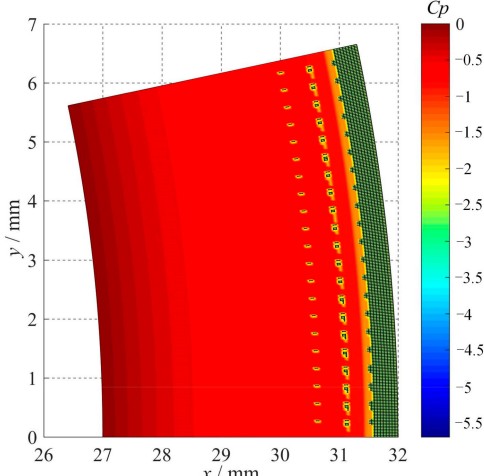

**Figure 12.** Pressure coefficient with encrypted texture rate.

## 3. Simulation Analysis

According to the numerical analysis results, cavitation occurred primarily inside the texture dimples, with the cavitation area concentrated at the friction disk's outer radius. The cavitation process was dynamically simulated using the fluid dynamics software CFX to further validate numerical analysis.

### 3.1. CFX Model Analysis

#### 3.1.1. Multiphase Models

The homogeneous flow model and the non-homogeneous flow model are the two most common models used in the selection of CFX, the former treats the gas–liquid two-phase

mixture as a homogeneous substance, and its flow parameters such as temperature and turbulence are taken as the weighted average of the two phases' corresponding parameters, so that each phase shares the same flow field and can quickly reach interphase equilibrium. In general, the homogeneous phase flow model is used in the cavitation process simulation, which is graphically represented in Figure 13 by treating non-uniform-sized bubbles as multiple small bubbles of equal size.

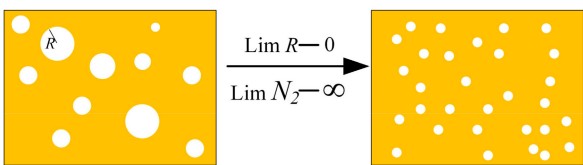

**Figure 13.** Schematic diagram of homogeneous model (R is radius of bubble; yellow area is liquid; bubbles are gas).

Furthermore, it is assumed that the parameter transport between the two phases is the same:

$$\varphi_\chi = \varphi \ (1 \leq \chi \leq N_P) \tag{23}$$

$$\frac{\partial}{\partial t}(\rho\varphi) + \nabla \bullet (\rho \boldsymbol{U}\varphi - \Gamma\varphi) = S_M \tag{24}$$

$$\rho = \sum_{\chi=1}^{N_P} \lambda_\chi \rho_\chi, \ \boldsymbol{U} = \frac{1}{\rho} \sum_{\chi=1}^{N_P} \lambda_\chi \rho_\chi \boldsymbol{U}_\chi, \ \Gamma = \sum_{\chi=1}^{N_P} \lambda_\chi \Gamma_\chi \tag{25}$$

where $\varphi$ is flow field's general scalar; $\chi$ is the different phases of the fluid, used as a subscript; $\rho$ is oil density, kg/m$^3$; $N_P$ is number of phases in fluid; $\lambda$ is volume fraction of each phase; $U$ is vector speed, m/s; $\Gamma$ is diffusion capacity between phases, kg/(m·s); $S_M$ is quality source, kg/(m$^3$·s).

### 3.1.2. Cavitation Models

There is a convective exchange of mass and volume fractions between the gas and liquid phases in cavitation flow [35], where the transport equation for the volume content of the gas phase is:

$$\frac{\partial(\lambda_v \rho_v)}{\partial t} + \nabla \cdot (\lambda_v \rho_v u) = \dot{m} \tag{26}$$

where $\lambda_v$ is volume fraction of gas phase; $\rho_v$ is gas density, kg/m$^3$; $u$ is gas vector velocity, m/s; $\dot{m}$ is the source term for evaporation and condensation.

The Rayleigh–Plesset equation describes bubble growth in liquids:

$$R_B \frac{d^2 R_B}{dt^2} + \frac{3}{2}\left(\frac{dR_B}{dt}\right)^2 + \frac{2\tau}{\rho_f R_B} = \frac{p_v - p}{\rho_f} \tag{27}$$

where $R_B$ is bubble radius, m; $p_v$ is the pressure in bubbles, Pa; $p$ is the pressure surrounding the bubble, Pa; $\rho_f$ is liquid density, kg/m$^3$; $\rho_v$ is gas density, kg/m$^3$; $\tau$ is the coefficient of surface tension between liquid and gas, N·m.

Equation (27) can be simplified by ignoring the second-order derivative term and the surface tension term:

$$\frac{dR_B}{dt} = \sqrt{\frac{2}{3}\frac{p_v - p}{\rho_f}} \tag{28}$$

The rate of change in bubble volume and mass can thus be calculated as follows:

$$\frac{dV_B}{dt} = \frac{d}{dt}\left(\frac{4\pi}{3}R_B^3\right) = 4\pi R_B^2 \sqrt{\frac{2}{3}\frac{p_v - p}{\rho_f}} \tag{29}$$

$$\frac{dm_B}{dt} = \rho_v \frac{dV_B}{dt} = 4\pi\rho_v R_B^2 \sqrt{\frac{2}{3}\frac{p_v - p}{\rho_f}} \tag{30}$$

If the quantity of bubbles in fluid is $N_B$, the fluid's vapor volume fraction can be demonstrated as:

$$\lambda_v = V_B N_B = \frac{4}{3}\pi R_B^3 N_B \tag{31}$$

Then, the total interphase mass transport rate between phases is:

$$\dot{m}_{fv} = F\frac{3\lambda_v\rho_v}{R_B}\sqrt{\frac{2}{3}\frac{p_v - p}{\rho_f}}\text{sgn}(p_v - p) \tag{32}$$

where $F$ is an empirical correction factor for the difference in evaporation or condensation rates, with evaporation being faster than condensation in general [40]. Because evaporation occurs primarily at the nucleation and causes a decrease in density there, the mass transport rate equation for the evaporation process must be rectified as:

$$\dot{m}_{fv} = F\frac{3\lambda_{nuc}(1 - \lambda_v)\rho_v}{R_B}\sqrt{\frac{2}{3}\frac{p_v - p}{\rho_f}}\text{sgn}(p_v - p) \tag{33}$$

where $\lambda_{nuc}$ is the volume fraction of nucleation centers.

When $p \le p_v$, evaporation occurs as described in Equation (28); when $p \ge p_v$, condensation occurs as described in Equation (27). The Rayleigh–Plesset cavitation model parameters in the CFX simulation are $R_B = 2$ μm, $\lambda_{nuc} = 5 \times 10^{-4}$, evaporation coefficient $F_{vap} = 50$, and condensation coefficient $F_{coud} = 0.01$.

### 3.2. Mesh Model Analysis

Due to the rotational periodicity of oil film and the time consumption in mesh division, the 1/30 oil film model is chosen for the analysis, and its parameters are shown in Table 2.

**Table 2.** Oil film model parameters.

| Parameters | Values |
|---|---|
| Inner radius $r_1$ (mm) | 27 |
| Outer radius $r_2$ (mm) | 32 |
| Center angle $\theta$ (°) | 12 |
| Default thickness $h_0$ (mm) | 0.1 |
| Default depth of texture $h_t$ (mm) | 0.0564 |

Figure 14 depicts the oil film flow field model when the rotating disk is textured with regular-arranged hexahedral micro-dimples. The textures are distributed 0.5 mm from the center of the adjacent textures along the radial direction and 1° from the center of the adjacent textures in circumferential direction. Figure 15 depicts the oil film model with a texture rate of 3.205%.

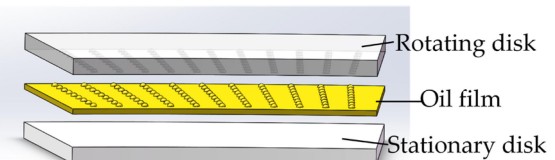

Rotating disk

Oil film

Stationary disk

**Figure 14.** Schematic diagram of friction pair structure.

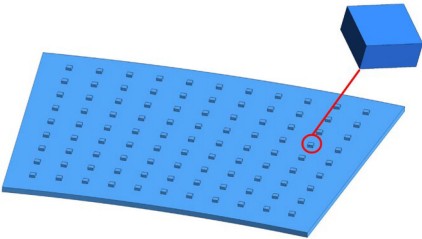

**Figure 15.** Schematic diagram of oil film with hexahedral texture.

Based on the literature review, cavitation is more likely to occur at high speeds, so the speed $n$ is set to 6000 rpm, the inlet pressure $p_0$ is set to 0.8 MPa, and the fluid's initial temperature is set to room temperature, $T_0$ = 293 K. Figure 16 depicts the boundary conditions for the simulation model.

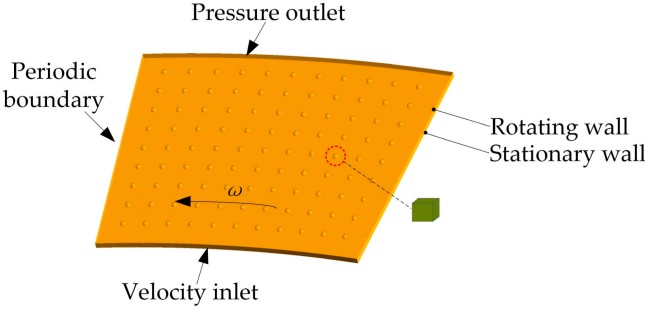

**Figure 16.** Boundary condition of flow field.

To eliminate the influence on the accuracy of the calculation results, most CFD simulation studies require mesh independence verification of the meshing. In comparison to cylindrical and hemispherical texture oil films, hexahedral texture meshing is relatively simple [41]. The cavitation volume fraction is used to determine computational accuracy and convergence under steady-state simulation conditions, and then to choose an appropriate meshing strategy. As in Table 3, Schemes 1–3 use different axially divided layers, Schemes 2–5 use different circumferentially divided layers, and Schemes 4–7 use different radially divided layers. Results show that increasing the number of axial layers excessively would cause simulation divergence, and the inaccuracy of the calculation results corresponding to different circumferential division of layers was smaller. Scheme 6 was chosen as the final meshing scheme to ensure calculation accuracy and save calculation time.

**Table 3.** Mesh segmentation strategy, number of grid cells, cavitation volume fraction.

| Scheme | Radial Layer | Circumferential Layer | | Axial Layer | Number of Grid Cells | Cavitation Volume Fraction |
| | | Texture Area | Non-Texture Area | | | |
| --- | --- | --- | --- | --- | --- | --- |
| 1 | 68 | 20 | 100 | 14 | 48,106 | 0.0004345 |
| 2 | 68 | 20 | 100 | 28 | 105,324 | 0.0003432 |
| 3 | 68 | 20 | 100 | 56 | 219,760 | Divergence |
| 4 | 68 | 30 | 120 | 28 | 137,484 | 0.0005012 |
| 5 | 68 | 40 | 160 | 28 | 184,384 | 0.0006212 |
| 6 | 102 | 30 | 120 | 28 | 207,252 | 0.0006417 |
| 7 | 136 | 30 | 120 | 28 | 277,020 | 0.0006419 |

### 3.3. Results of Simulation Analysis

### 3.3.1. Effect of Texture Rate on Cavitation

In the literature summary, cavitation rarely happens on absolutely smooth surfaces. The effect of different texture rates is investigated in this subsection. For the transient

simulation of cavitation, the default texture rate was 3.205%, which increased exponentially, while a non-texture oil film was used as a control group. First, when the rate was 0, the cavitation volume fraction was 0, indicating that no cavitation occurred on the smooth surface. When the rate was 3.205%, the fraction was $9.66 \times 10^{-4}$, and it stabilized at 0.00151 and 0.00297 when the texture rate was increased to 6.41% and 12.917%, respectively. Cavitation area expands as the texture rate increases. However, it should be noted that this conclusion applies only to the hexahedral texture studied in this research and not necessarily to other types of texture. Considering that the coverage of the hexahedral texture is proportional to the cavitation strength, when processing the hexahedral texture for hydraulic machinery such as wet clutches or mechanical seals, an optimal texture rate must be considered in order to balance the optimal film carrying capacity of the oil with a low cavitation effect.

As shown in Figure 17, the cavitation volume fraction increased briefly during the simulation cycle before returning to a stable state. Although the final values varied, they all peaked within 0.015 s. In Figure 18, When the rate is 3.205%, cavitation occurs only inside the texture dimples, the spacing between adjacent textures is greater, and the hexahedral dimples resemble isolated rough elements. When oil flows into the dimples, the texture roots are subjected to greater shearing action, and cavitation is more likely to occur at the textures' downstream low-pressure position. As shown in Figures 19 and 20, as the texture rate increases, cavitation is no longer limited to the interior of the texture dimples, and the higher density of texture dimples increases the intensity of the boundary layer turbulence, causing cavitation to occur in the non- texture region at the outer warp.

The CFX simulations support that the simplifying assumptions made in the Reynolds equation model, computed by a finite difference algorithm, are a sufficiently good approximation. Comparing Figures 11a and 17, the cavitation volume fraction is 0 when texture rate is 0. Comparing Figures 11b and 18, when the texture rate is low, the cavitation vapor preferentially appears inside the texture dimples at the outer diameter, which proves that cavitation is more likely to occur at larger angular velocities. Comparing Figures 12 and 18, it is found that after the texture rate increases, the cavitation vapor is still concentrated at the outer diameter. The cavitation vapor is still concentrated at the outer diameter but is no longer confined to the texture dimples and cavitation vapor starts to appear in the non-texture region.

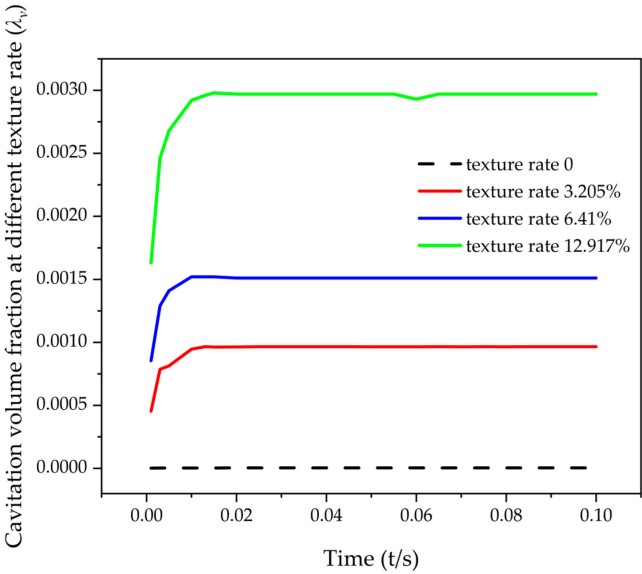

**Figure 17.** Cavitation volume fraction at different texture rates.

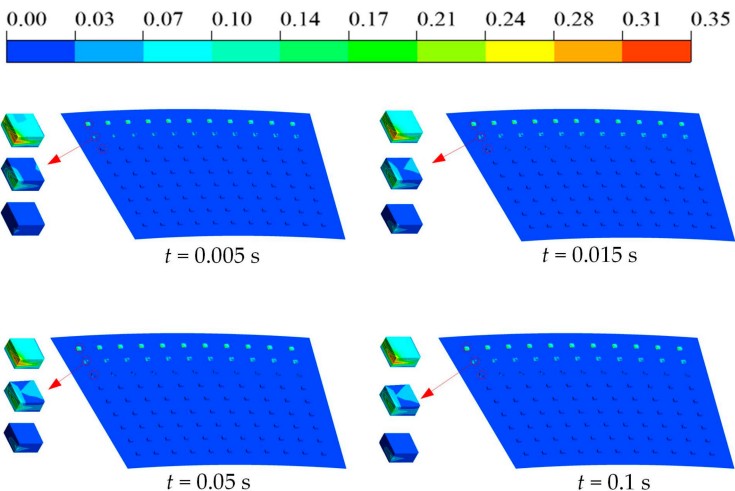

**Figure 18.** Cavitation morphology change with hexahedral texture rate 3.205%.

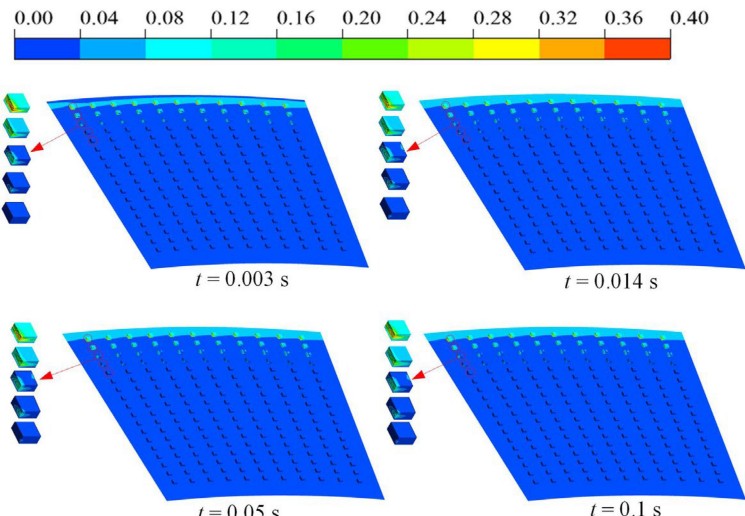

**Figure 19.** Cavitation morphology change with hexahedral texture rate 6.41%.

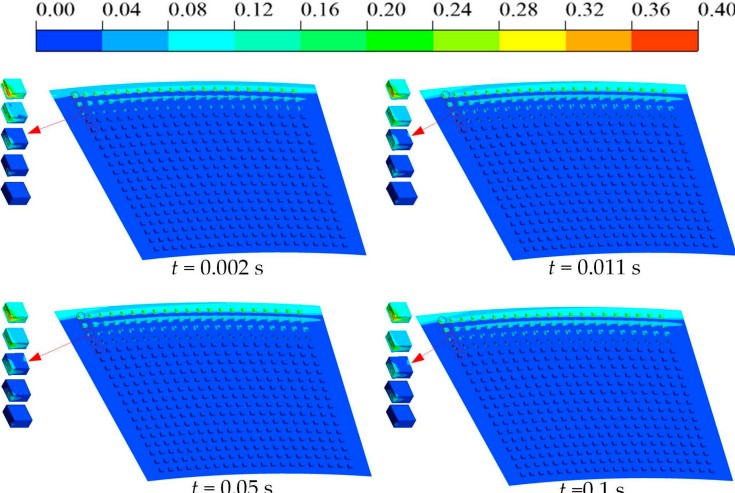

**Figure 20.** Cavitation morphology change with hexahedral texture rate 12.917%.

### 3.3.2. Effect of Cavitation on Velocity

The oil film with a texture rate of 6.41% was chosen to analyze the flow and cavitation vapor movement in hexahedral dimple. The local area at the upper left corner of the oil film is enlarged in Figure 21, where (a1), (a2), and (a3) are the oil velocity vector diagrams at different times, and we can see that the oil flows smoothly in the non-texture region and induces turbulence when it passes through the texture boundary. The vapor vector diagrams are represented by (b1), (b2), and (b3), respectively. Because the arrows of the cavitation vapor and oil remain essentially the same, we can assume that the vapor originated from the oil's phase change.

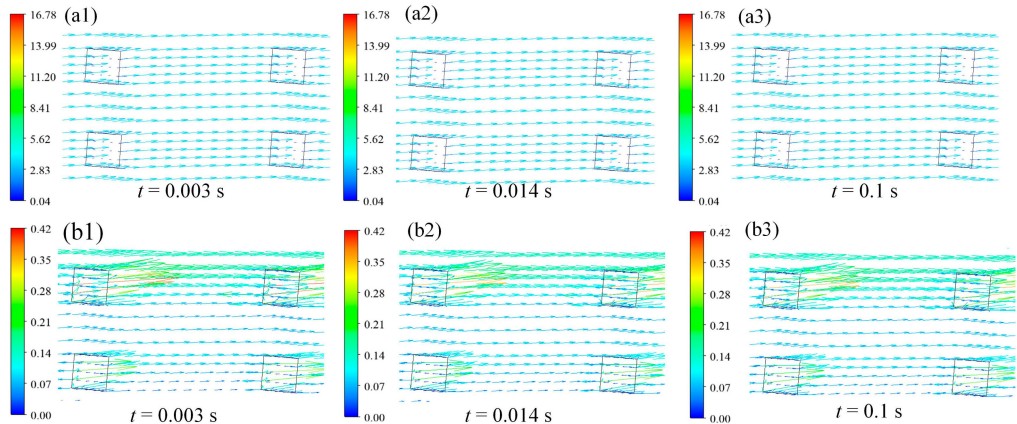

**Figure 21.** Vector diagram of local oil flow velocity in oil films with different times ((**a1**) is velocity vectors of oil when $t = 0.003$ s; (**a2**) is velocity vectors of oil when $t = 0.014$ s; (**a3**) is velocity vectors of oil when $t = 0.1$ s; (**b1**) is velocity vectors of vapor when $t = 0.003$ s; (**b2**) is velocity vectors of vapor when $t = 0.014$ s; (**b3**) is velocity vectors of vapor when $t = 0.1$ s).

The fluid velocity streamlines on the internal cross section of a single texture dimple are shown in Figure 22. Where (a1), (a2), (a3) are velocity streamlines of oil at different times, correspondingly, (b1), (b2), (b3) are velocity streamlines of vapor. In the non-texture region, oil flows relatively smoothly, and the hexahedral structure has a more pronounced effect on oil retention. When flowing into the hexahedral dimple, two large vortices are generated, and the lowest pressure often appears in the vortex center, so it is where cavitation occurs initially. The relationship between cavitation and pressure will be investigated further.

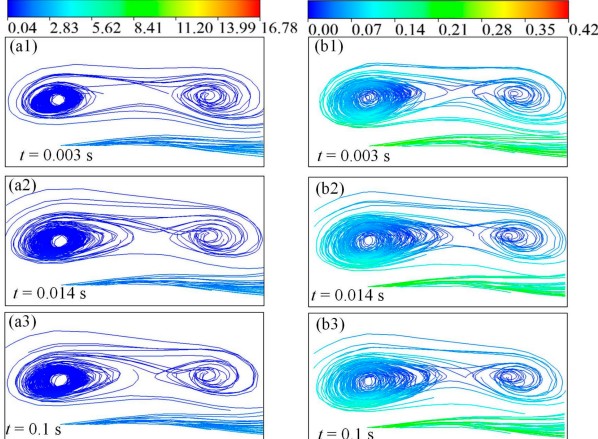

**Figure 22.** Velocity streamlines of the section inside the hexahedral texture dimple ((**a1**) is velocity streamlines of oil when $t = 0.003$ s; (**a2**) is velocity streamlines of oil when $t = 0.014$ s; (**a3**) is velocity streamlines of oil when $t = 0.1$ s; (**b1**) is velocity streamlines of vapor when $t = 0.003$ s; (**b2**) is velocity streamlines of vapor when $t = 0.014$ s; (**b3**) is velocity streamlines of vapor when $t = 0.1$ s).

### 3.3.3. Effect of Texture Depth on Cavitation

To investigate the effect of texture depth on cavitation, $h_t$ was set to 0.05 mm, 0.1 mm, and 0.15 mm, respectively. In Figure 23, the cavitation degree decreases as the texture depth increases, and Newton's law of internal friction states that the fluid in the dimple is subjected to stronger shear at smaller depths, making the oil film more prone to cavitation. As the depth increases, the space inside the dimple expands, and the oil in the texture area is subjected to less shear stress, resulting in a gradual decrease in the cavitation area.

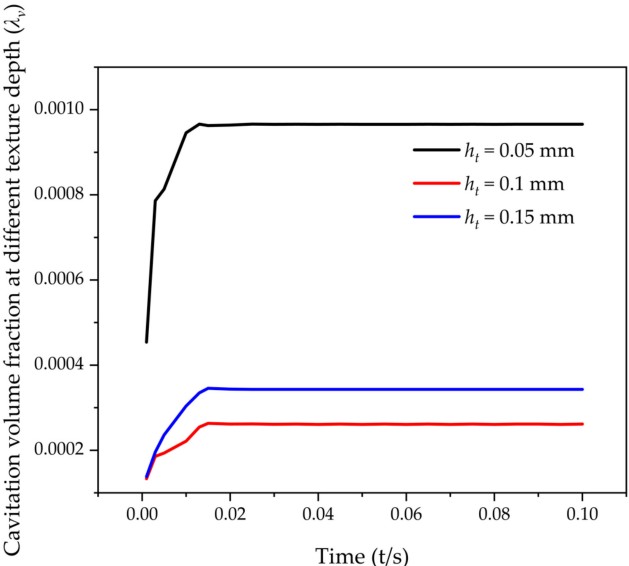

**Figure 23.** Effect of texture depth on oil film cavitation.

So, according to Figure 24, cavitation occurs primarily at the root of the oil film microconvex body, i.e., at the entrance of the texture dimple.

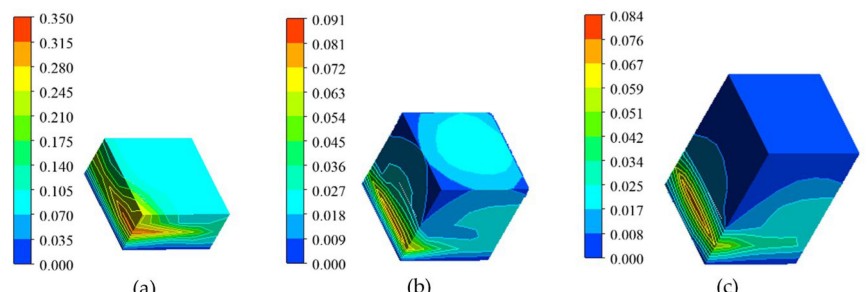

**Figure 24.** Cavitation volume distribution in texture dimples at different depths ((**a**) is cavitaion contour when texture depth is 0.05 mm; (**b**) is cavitaion contour when texture depth is 0.1 mm; (**c**) is cavitaion contour when texture depth is 0.15 mm).

### 3.3.4. Pressure Analysis in Cavitation Area

Figure 25 depicts the flow field pressure distribution at $n$ = 6000 rpm. The texture dimples affect the pressure distribution, but the oil pressure distributed consistently overall except for the oil near the dimples, and the cavitation vapor appears at the same location as the low-pressure area.

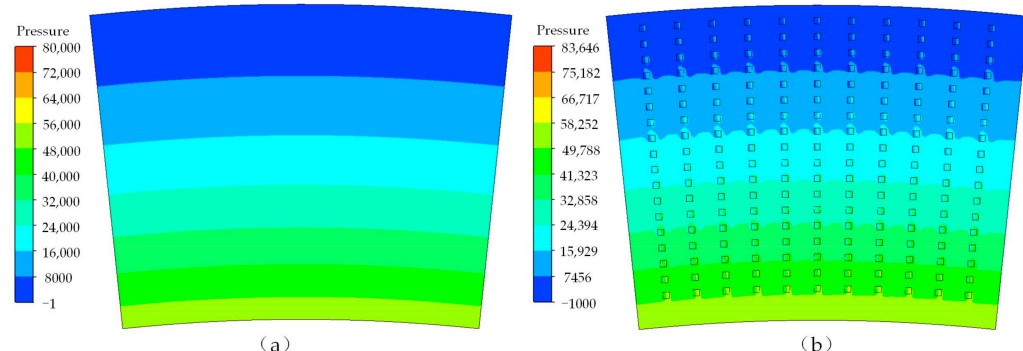

**Figure 25.** Pressure contour of oil film ((**a**) is pressure distribution of oil film with textures; (**b**) is pressure distribution of oil film without textures).

Figure 26 depicts the radial distribution of oil pressure on the upper surface of oil film, with overall pressure decreasing along the radial direction; $\theta = 6°$ is in the texture region, and the oil flow through the texture dimple causes a significant pressure pulsation, where $\theta = 5.5°$ is in the non-texture region, and the pressure curve is relatively smooth. As a consequence, pressure turbulence caused by the weaving structure is the primary cause of cavitation.

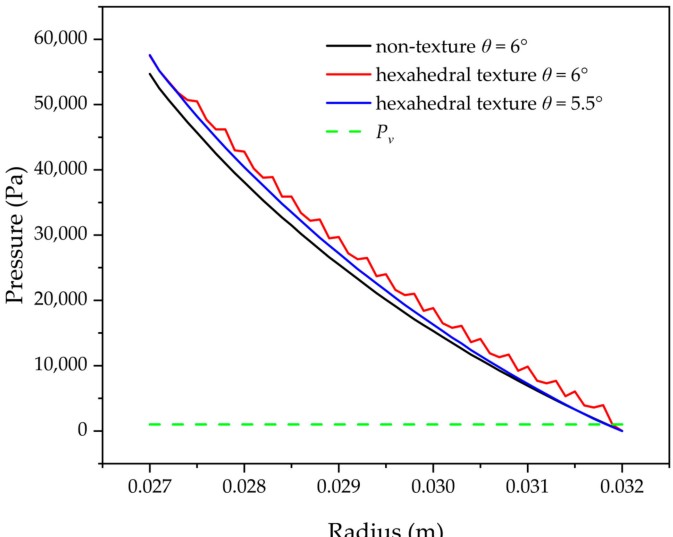

**Figure 26.** Radial pressure distribution of oil film.

Conversely, whether cavitation affects pressure requires further investigation. To statistically measure the pressure distribution inside a single texture, we ran simulations under two conditions: one considering the cavitation effect and one disregarding the cavitation effect [42,43]. Figure 27 depicts the pressure distribution inside a single texture dimple at various rotational speeds while accounting for cavitation effect (0.9–1.1° for the texture region, 0.8–0.9° and 1.1–1.2° for the non-texture region). The oil is strongly squeezed near the texture root due to the strong shearing effect and barrier effect in dimple, resulting in a significant decrease in the low-pressure value and a significant increase in the high-pressure value.

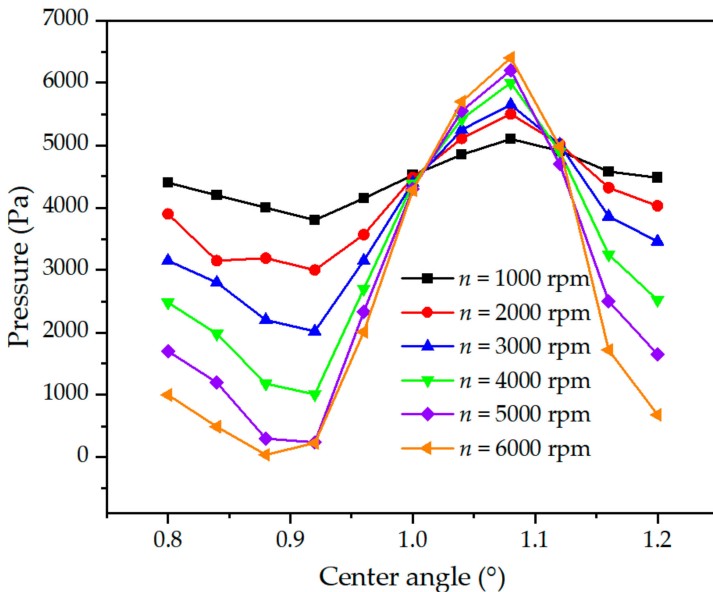

**Figure 27.** Pressure distribution in texture dimple with cavitation effect at different rotational speeds.

Figure 28 depicts the pressure distribution in the texture dimple at various rotational speeds while ignoring the cavitation effect. The pressure distribution in the non-texture region has the same pattern regardless of cavitation effect, whereas the pressure distribution in the texture region has the same pattern at low rotational speeds, i.e., when cavitation does not occur. As the speed increases, particularly at $n = 6000$ rpm, the low-pressure region has a lower peak pressure of $-530$ Pa when the cavitation effect is ignored, whereas the low-pressure peak increases when the cavitation effect is considered, implying that the cavitation effect partially enhances the fluid dynamic pressure effect.

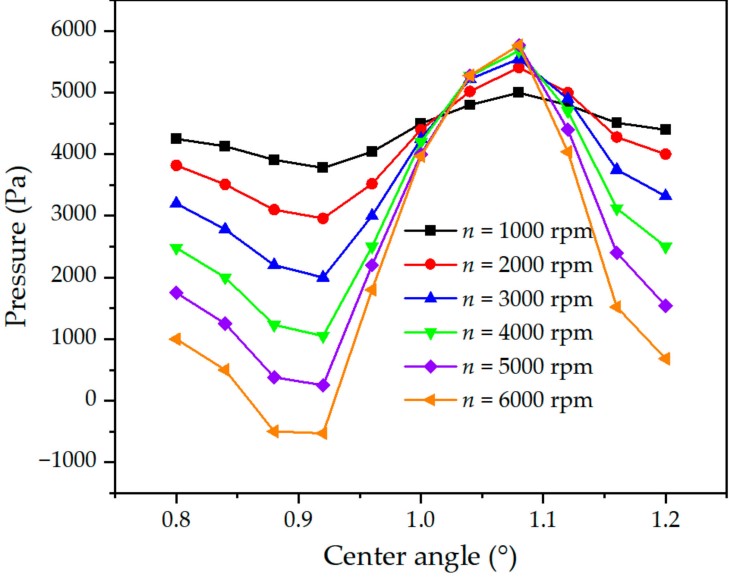

**Figure 28.** Pressure distribution in texture dimple without cavitation effect at different rotational speeds.

## 4. Experimental Analysis

Although consistent conclusions were reached in numerical calculations and simulations, whether the theoretical analysis matches the results in actual practical engineering applications needs to be investigated further using experiments.

### 4.1. Experiment Design

The design of a single pair of friction pair visualization test systems, the principle of which is shown in Figure 29, consists of three parts: transparent friction device, hydraulic system, and data acquisition system to verify the validity of the theoretical analysis in practical engineering applications.

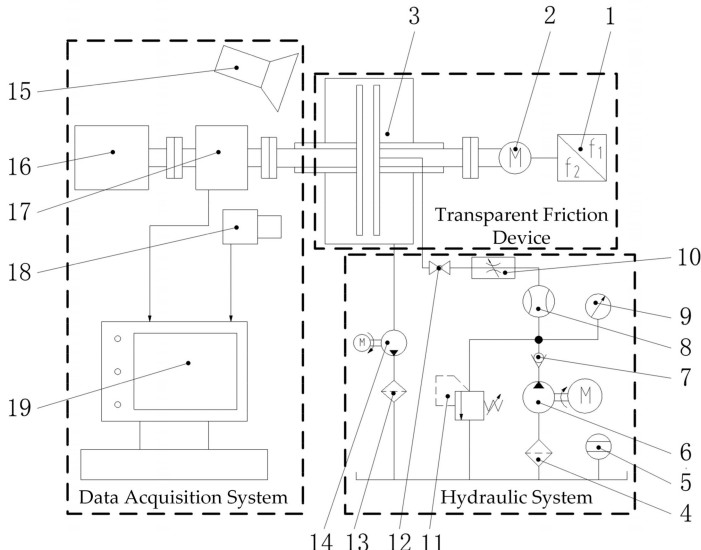

**Figure 29.** Transparent test system (1 = Inverter; 2 = Three-phase asynchronous motor; 3 = Transparent host; 4 = Suction filter; 5 = Liquid level meter; 6 = Quantitative gear pump; 7 = Check valve; 8 = Flow meter; 9 = Pressure gauge; 10 = Speed control valve; 11 = Relief valve; 12 = Guide-off valve; 13 = Return filter; 14 = Triangular cycloidal pump; 15 = Fill light 16 = Holding device; 17 = Torque tachometer; 18 = High-speed camera; 19 = Computer).

A laser-marking machine was used to texture the surface of the friction disk in order to analyze the phenomenon of oil film cavitation, some of the laser marking machine's parameters were set as shown in Table 4, and the textured friction disk was processed as shown in Figure 30.

**Table 4.** Laser processing parameters.

| Parameters | Value | Parameters | Value |
|---|---|---|---|
| Output power | 70% | Laser wavelength | $1064 \pm 5$ nm |
| Laser processing speed | 500 mm/s | Processing speed | 1000 mm/s |
| Spot diameter | 0.05 mm | Repetition accuracy | $\pm 0.003$ nm |

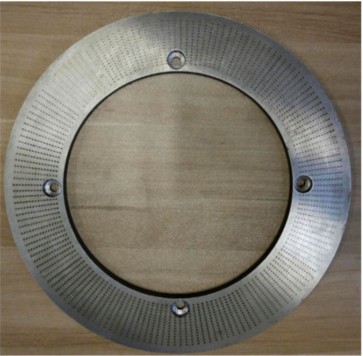

**Figure 30.** Textured surface of a friction disk.

A transparent experimental table, as shown in Figure 31, was installed according to the experimental system's design. The experiment's core component is the use of high-speed cameras to capture the oil flow and cavitation development patterns under various working conditions. The high-speed camera model is Thousand Eyes Wolf 5F01 (Wuxi Mu Xin Wisdom Information Technology Co., Wuxi, China), its resolution is 2320 × 1720, and shooting speed is 500~52,000 frames per second.

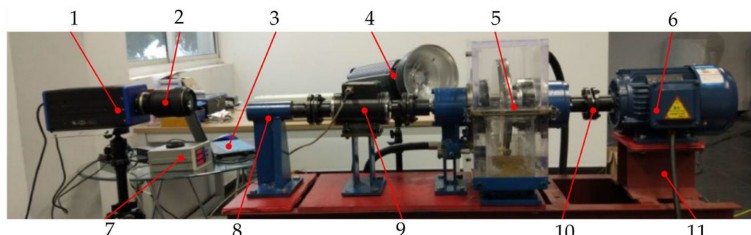

**Figure 31.** Transparent test platform (1 = high-speed camera; 2 = lens; 3 = computer; 4 = fill light; 5 = host; 6 = motor; 7 = data collection unit; 8 = magnet powder brake; 9 = torque sensor; 10 = coupling; 11 = base).

*4.2. Experiment Results*

The motor's speed was gradually increased from 300 rpm to 1200 rpm using a frequency converter, and finally 350 rpm, 550 rpm, and 1050 rpm were chosen to correspond to low, medium, and high speeds, respectively, based on the captured images. It should be noted that the previous paper argued that cavitation is related to angular velocity, the theoretical analysis part used a smaller size model to simplify the calculation, and the inner and outer diameters of the friction disk in the experiment are much larger than those in the theoretical part.

To demonstrate that rough elements can cause cavitation, a non-texture friction disk was first installed as a control experiment on a transparent experimental bench. Figure 32a–c shows the surface cavitation pattern of the non-texture friction disk under different speed conditions, and no cavitation bubbles are generated in the field of view under all three speed conditions, demonstrating the same pattern as Figures 11 and 17. It is once again demonstrated that cavitation does not occur on smooth or near-smooth surfaces.

Following that, the non-texture friction disk was replaced with a 6% texture friction disk, and as shown in Figure 32d, there were no cavitation bubbles at 350 rpm, indicating that there was no cavitation at low speed. As shown in Figure 32e, cavitation bubbles appeared first at the outer diameter at 550 rpm. As shown in Figure 32f, multiple narrow cavitation bubbles appeared in the view field at 1050 rpm, indicating.

The results of modeling calculations, CFX simulations, and experiments are consistent in several ways: cavitation does not occur on smooth surfaces, cavitation preferentially occurs on the area around the outer diameter (larger angular velocity), and increasing rotational speed enlarges the cavitation region. However, the texture structure in this experiment is not a regular hexahedron, the experiments cannot capture the cavitation generation process microscopically, and the modeled fluid is an ideal medium free of impurities; all of these limitations result in non-conformity between the experiments and the modeling, and this will be the focus of future research.

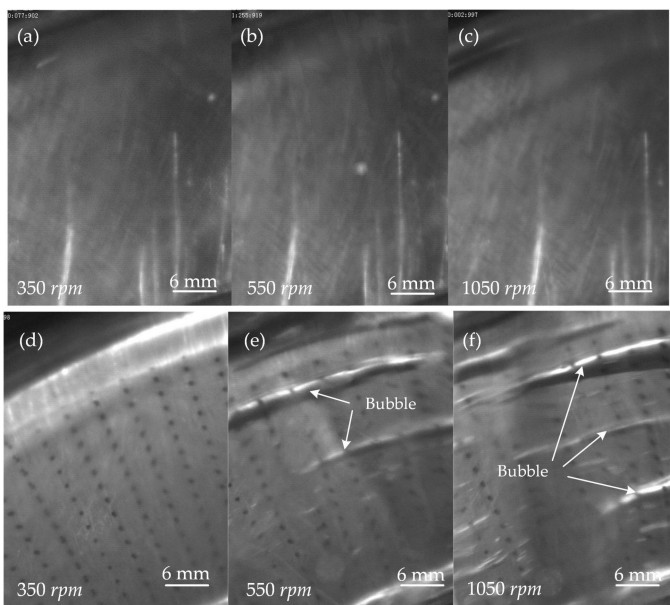

**Figure 32.** Cavitation morphology on non-textured friction disk and textured friction disk ((**a**) is cavitation morphology on non-textured friction disk when speed is 350 rpm; (**b**) is cavitation morphology on non-textured friction disk when speed is 550 rpm; (**c**) is cavitation morphology on non-textured friction disk when speed is 1050 rpm; (**d**) is cavitation morphology on textured friction disk when speed is 350 rpm; (**e**) is cavitation morphology on textured friction disk when speed is 550 rpm; (**f**) is cavitation morphology on textured friction disk when speed is 1050 rpm; The arrows in (**c**) and (**d**) indicate the location of the cavitation bubbles).

## 5. Conclusions

The oil film between the friction disks processed with arranged hexahedral texture dimples is the research object in this study to analyze the oil flow behavior in the rotating disk system. The velocity and pressure of the oil film are solved using a finite difference algorithm, and the pressure coefficient of the flow field is calculated to predict the cavitation incipient position under different texture distribution densities. To simulate the cavitation process and verify the numerical analysis, a 3D model was created and calculated in CFX software (ANSYS, Canonsburg, PA, USA). The study results can be applied to cavitation prevention in instruments, such as hydro-viscous clutches, wet clutches, disk turbines, and disk motor cooling systems. This paper's research and summary are as follows.

The mathematical model of the oil film is established based on Reynolds equation, and the oil film's velocity, pressure, and pressure coefficient are solved by finite difference algorithm. The results show that the hexahedral texture dimples change the velocity and pressure in the flow field. The oil's radial velocity is greater in the texture dimple than in the non-texture region. The cavitation area is confined to dimples at the outer diameter at lower texture densities. Cavitation occurs in the non-texture area at outer warp as the density of the texture distribution increases.

ANSYS CFX was used to simulate the cavitation flow between the rotating disk system. The cavitation locations are found to be consistent with the numerical analysis' conclusions. When texture rate is low, a single isolated texture dimple causes a localized disturbance that affect the overall stability slightly. Experiments were used to confirm the above results and reach a consistent conclusion. The disturbance intensity in the boundary layer increases as the number of textures increases, resulting in cavitation in the non-texture region. It should be noted, however, that this is only applicable to the hexahedral texture. The cavitation performance of other texture types needs to be investigated further. The greater the texture depth, the smaller the cavitation volume fraction.

Analysis of velocities of the vapor and liquid phases flowing through the texture cross-section and the flow inside the hexahedral texture dimple revealed that the flow

direction and flow pattern between the vapor and liquid phases remained essentially the same. Cavitation occurs when there is a low pressure and a high speed. Cavitation and pressure have an effect on one another. On the one hand, cavitation is more likely to occur in low-pressure areas because low pressure induces cavitation. Cavitation, on the other hand, results in a smaller low-pressure peak and a negative pressure. As a result, the cavitation effect is thought to enhance the dynamic pressure effect.

## 6. Patents

The patent number is CN201810435987.8.

**Author Contributions:** Conceptualization, J.S.; methodology, J.S.; software, J.S.; validation, L.C., B.Z. and P.Q.; formal analysis, J.S.; investigation, J.S.; resources, J.S.; data curation, H.H.; writing—original draft preparation, J.S.; writing—review and editing, J.S.; visualization, J.S.; supervision, L.C.; project administration, B.Z.; funding acquisition, B.Z. and P.Q. All authors have read and agreed to the published version of the manuscript.

**Funding:** This research was funded by the National Natural Science Foundation of China (51805215, 51605194), in part by the Starting Foundation of Jiangsu University Advanced Talent (14JDG048), in part by the Open Foundation of the State Key Laboratory of Fluid Power and Mechatronic Systems (GZKF-201819), and in part by the Scientific Research Foundation of Tangshan Normal University of China (Grant No. 2021A02).

**Institutional Review Board Statement:** The study did not require ethical approval.

**Informed Consent Statement:** The study is not involving humans.

**Data Availability Statement:** The data that support the findings in this study are available from the author J.S. upon reasonable request.

**Acknowledgments:** In addition to the efforts of the co-authors, I need to thank the School of Mechanical Engineering of Jiangsu University for providing the experimental apparatus and experimental materials. As well as colleagues from the Institute of Mechatronic and Hydraulic Control for their help in the experiments.

**Conflicts of Interest:** The authors declare no conflict of interest.

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
