# Peer review of "Comprehensive Investigations on Fluid Flow and Cavitation Characteristics in Rotating Disk System"

_applsci, doi:10.3390/app122312303_

Round 1

Reviewer 1 Report

The authors conducted a numerical investigation of the fluid flow (Newtonian) behavior between rotating disk systems. The paper looks great, however, I have a few questions subjected to this work.

1)   Based on the abstract: "cavitation process is simulated in CFX and eventually verify the correctness of the numerical analysis". Please elaborate in the discussion part on how this process validates the model and method.

2) The conclusion part is too long. Please make it interesting by putting them in rebuttal points.

Author Response

Dear Reviewer,

Thank you very much for your comments. In order to clearly answer your questions, I have recorded the modification process in the attachment. Thank you for your time.

Kind regards

Sun Junyu

Reviewer 2 Report

This paper is novel and original by construction.   An original finite difference scheme algorithm is developed.  A CFX model for hydromechanical cavitation for an oil over a rotating disc (same configuration) is also developed.   Both are explored.

What seems to be the case, however, is that the basis of the two models is different, with the finite difference model making simplified assumptions.  The rationale for why there are two models is not given, other than validation by comparison.   Comparing two models identifies the parametric regime where the two models are in good agreement, but allows no real confidence that either is a correct prediction of reality.   The essential feature of grid independence is shown for the CFX model, so that it is a sufficiently accurate approximation to at least the prediction of the volume fraction of cavitation.   But is there any way to benchmark the predictions against some experimental study?

Overall, I cannot recommend for publication until the basis for why there are two models, to the extent that they are different is explained, and whether or not there is any benchmarking by experiment is discussed.  

Some incorrect points:

1.  Column coordinates probably means cylindrical (polar) coordinates.

2.  Circumferential velocity probably means angular velocity.  Or is it the angular velocity at the disc edge?

3.  "refers to introduce" should be "refers to the introduction of"

4.  Conventionally, cavitation is a phase change from liquid TO VAPOR or TO GAS via release of dissolved gas.

5. "The key incentive of cavitation" ...  human behavior (incentive) cannot be ascribed to inanimate materials.   Impetus is a better word choice, but the sense is better served with "The key impetus FOR cavitation is the formation of the cavitation nucleus ..."    Tim Leighton's classic book, The Acoustic Bubble, describes the essential component for low power nucleation are impurities in the water that form nucleation sites.  Energetics analysis of the cohesion force of water show that much higher power density is needed unless there are contaminants.   So is sigma (equation 1) consistent with nucleation with or without contaminants?   Equation 1 does not fit with the earlier assertion that nucleation is the key impetus, but rather rarefaction is the key impetus.   Both are surely necessary.

6.  "due to cavitation bubbles created"  -- no apostrophe

7. evaporation being faster than condensation in general -- do you have a reference for this, or at least more description.  Is there an implicit assumption that evaporation is caused by nucleation only on the surface (heterogeneous nucleation)?   Is it assumed condensation can occur anywhere along the bubble interface?  Does the whole paper assume that there is no homogeneous nucleation (in the bulk liquid)?

8.  Mesh number is a strange terminology.   It does not seem to be defined, so one would think that the number of grid cells / volume elements is the  intended concept.   It is also referred to as mesh independence or grid independence, not irrelevance!

Author Response

Dear Reviewer,

Thank you very much for your professional suggestions. Based on your comments, I learned a lot while revising my article. I was very shocked and moved to see Incorrect point 5. This will motivate me to be more rigorous and attentive in my future research. In response to your question about the practicality of the model in numerical calculations and simulations. I have explained in the annexes, and I have added an experimental study section to the article to better verify the engineering application value of this study.

Corresponding Author: Junyu Sun

Round 2

Reviewer 2 Report

The authors have addressed all of my concerns in the original review.

One minor point, however, is loose terminology around line 440:

"The CFX simulation process verifies the correctness of Reynolds equation differential differentiation algorithm ..."

The CFX simulations supports that the simplifying assumptions made in the Reynolds equation model, computed by a finite difference algorithm, are a sufficiently good approximation.

I would call section 4 "Experimental".

Finally --

The point of this section seems to be that rough texture results in cavitation, which is a gross agreement with the modelling and simulation.  Detailed agreement is not illustrated, which would validate the modelling.  The authors should at least briefly discuss the extent to which the experimental analysis lends confidence to this gross prediction of the model.  

One modelling purpose is to predict emergent behavior of a complex system, so that it can be understood.  So this gross emergent feature being predicted by the modelling is at least consistent.  How consistent is not assessed.   It is okay for a modelling paper to state that detailed comparison between the modelling predictions and experimental analysis is future work beyond the scope of the paper.  But it should be self critical enough to point out the limitations of the work.

Otherwise, I am satisfied that the paper should be accepted.

Author Response

Dear Reviewer,

Thank you very much for your time and kind help. The consistency between the experiment and the model is discussed at the end of the article, and the flaws of this paper's study are presented. It is true that being self-critical and confronting the study's flaws takes courage in order to make a breakthrough.

Kind regards.
